# Quality of Life in Metabolic Syndrome Patients Based on the Risk of Obstructive Sleep Apnea

**DOI:** 10.3390/bs14020127

**Published:** 2024-02-09

**Authors:** Taehui Kim

**Affiliations:** Department of Nursing Science, Joongbu University, Chungnam 32713, Republic of Korea; skyibe@joongbu.ac.kr

**Keywords:** metabolic syndrome, sleep apnea, obstructive, quality of life

## Abstract

Despite the impact of metabolic syndrome (MetS) and obstructive sleep apnea (OSA) on a sizeable proportion of the global population, the difference in the quality of life (QoL) between a group without risk factors for OSA and a group with risk factors for OSA among individuals with MetS is currently unclear. This study aimed to identify the determinants of QoL in patients with MetS with and without OSA risk factors and to analyze differences between these two groups. Data were extracted from the 2020 Korea National Health and Nutrition Examination Survey (KNHANES). The Rao–Scott χ^2^ test was performed to evaluate differences in baseline characteristics based on OSA risk factors. A *t*-test was performed to evaluate differences in the baseline QoL, and linear regression analysis was performed to identify the effect on the QoL of the two groups. The factors affecting QoL in the low-risk group included age, education level, and depression. The factors affecting QoL in the high-risk group were physical activity and depression. These results suggest that nursing interventions should be devised according to patients’ characteristics to help improve their QoL.

## 1. Introduction

According to a report based on data extracted from the National Health and Nutrition Examination Survey (NHANES), the prevalence of metabolic syndrome (MetS) in the United States is 34.7% [1]. The recorded prevalence rates of MetS are 27.9% in Korea in 2018, 28.4% in Indonesia in 2006, 34.3% in Malaysia in 2008, and 25.5% in Taiwan in 2005–2008 [2]. These rates indicate that the prevalence of MetS among the global population is in excess of 20%.

MetS refers to a combination of three or more of the following conditions: abdominal obesity, elevated triglycerides, decreased high-density lipoprotein cholesterol, elevated blood pressure, and insulin resistance [3]. Consequently, MetS increases the risk of a large number of conditions including stroke, diabetes mellitus (DM), chronic renal failure, myocardial infarction (MI), and angina pectoris (AP), all of which increase mortality [4], with 87.8% of the major mortality rate in high-income countries notably coming from noncommunicable diseases, such as heart disease and stroke [5].

Obstructive sleep apnea (OSA) occurs in 1–2% of the population and is a common condition in 20–57% of middle-aged men and 10–41% of middle-aged women [6]. When OSA was investigated in the general population, 41–58% of individuals showed a score of ≥5 in the apnea–hypopnea index [6]. Apnea causes suffocation and arousal in OSA patients during sleep [7]. This changes normal physiology, causing cardiovascular and metabolic diseases, such as systemic hypertension, diabetes, coronary artery disease, stroke, and increased cardiovascular mortality, and affecting mood disorders, including depression [8,9]. OSA is also closely associated with abdominal obesity, which has greater relevance after the COVID-19 pandemic increased the prevalence of obesity in individuals of all ages [5]. Furthermore, moderate-to-severe OSA increases the risk of developing MetS by 2.6, cardiovascular disease by 2.48, and stroke by 2.02 times [10,11]. 

Previous studies have shown an association between MetS and OSA [8,12]. OSA is also associated with hypertension, stroke, heart failure, and coronary artery disease, all of which are associated with MetS [4,9]. Type 2 diabetes serves as a diagnostic factor in conjunction with MetS and OSA [9,12]. Additionally, age, gender, hypertension, and dyslipidemia are common risk factors associated with MetS and OSA [9,12].

What is the quality of life (QoL) of individuals with both MetS and OSA? The concept of QoL is complex, and various approaches related to personal relationships, work and life satisfaction, feelings about life, disposition, and individuals’ feelings about their situations should be examined to investigate QoL appropriately [13]. QoL represents individuals’ perception of their position in life according to the culture and value systems in which they live and in relation to their own goals, expectations, standards, and concerns [14]. The QoL of patients with MetS is known to be generally low, and the factors affecting it are depression, body mass index (BMI), stress, lifestyle, and health status [15,16]. The QoL of patients with OSA is similar to that of patients with MetS and is lower than that of the general public [17]. Moreover, the development of MetS is also associated with age, sex, BMI, smoking, alcohol consumption, marriage, and depression [18]. However, the abovementioned studies are non-contemporary, and the current level of research remains insufficient. In addition, few studies have compared the QoL of patients with both MetS and OSA to those with only MetS. The QoL of patients with both MetS and OSA and that of patients with MetS only may be similar, or significant differences may exist between both groups.

Therefore, this study was conducted to investigate factors influencing the QoL of patients with MetS, comparing these factors in two groups—one at risk of OSA and the other not at risk.

## 2. Materials and Methods

### 2.1. Participants

This study used data from the 2020 Korea National Health and Nutrition Examination Survey (KNHANES), a nationwide cross-sectional survey conducted annually by the Korean Disease Control and Prevention Agency (KDCA). The study participants are representatives of the non-institutionalized population in Korea. The sampling plan followed a multi-stage clustered probability design. In total, 7359 surveys were conducted in 2020. Since 2020, the KDCA has used STOP-Bang as a screening tool for OSA to investigate the sleep health of individuals aged ≥ 40 years. The study sample included 4165 adults aged 40 and above, with 1589 of them having MetS, while 2576 did not have MetS. Among the 1589 participants with MetS, 748 were categorized as having low-risk factors for OSA, while 841 were classified as having high-risk factors for OSA (Figure 1). 

### 2.2. Variables

#### 2.2.1. QoL

The KNHANES measured the QoL using EuroQoL-5Dimension (EQ-5D). The KDCA’s use of EQ-5D was approved by the EuroQoL Group. EuroQoL-5 Dimension-3L consists of mobility, self-care, usual activities, pain or discomfort, and anxiety or depression [19]. It comprises three points (1 = no problem; 2 = some problems; and 3 = extreme problems), and is weighted by the EQ-5D index—the higher the score, the higher the QoL [19].

#### 2.2.2. MetS

For a MetS diagnosis, modified versions of the guidelines from the National Cholesterol Education Program Adult Treatment Panel III of the International Diabetes Federation and American Heart Association, and from the National Heart, Lung, and Blood Institute were adopted [3]. The criteria used to define abdominal obesity was the Korean Society’s definition [15]. MetS is defined in this study as the presence of three or more of the following components [3]:(1)Abdominal obesity: A waist circumference of ≥90 cm for men, and of ≥85 cm for women;(2)Hypertriglyceridemia: A triglyceride concentration of ≥150 mg/dL or, the reception of a specific treatment for this lipid abnormality;(3)High-density lipoprotein cholesterol: A serum high-density lipoprotein cholesterol concentration of <40 mg/dL for men, and of <50 mg/dL for women, or the reception of a specific treatment for lipid abnormality;(4)High blood pressure: A systolic blood pressure ≥ 130 mmHg and a diastolic blood pressure ≥ 85 mmHg, or the reception of treatment with antihypertensive agents;(5)High fasting glucose: A fasting serum glucose level of ≥100 mg/dL or the reception of treatment through antidiabetic medication

#### 2.2.3. OSA

The KDCA used the STOP-Bang questionnaire to investigate obstructive sleep apnea in adults aged 40 and older [20]. The risk of OSA was determined through the STOP-Bang score (Table 1).

According to the STOP-BANG classification guidelines, individuals are categorized into three groups. (i) low risk: answer yes to 0–2 questions; (ii) intermediate risk: answer yes to 3–4 questions; (iii) high risk: answer yes to 5–8 questions. In this study, individuals who responded with fewer than two ‘yes’ answers were classified into the low-risk group, while those who responded with three or more ‘yes’ answers were categorized into the high-risk group. 

#### 2.2.4. Demographic Characteristics

The following variables were selected: age, sex, education, occupation, habitation with family members, and household income. The highest level of education was graduation. Age was classified as 10 years, 40 years, or older. Household income was classified into four quartiles of sample households and the sample population: “high”, “middle-high”, “middle-low”, and “low”. They were then reclassified as “high”, “middle”, or “low”. A question about the participants’ economic activity was used to measure their occupation status. Employed participants’ occupation status was classified as “yes”, whereas that of unemployed participants was classified as “no”. Regarding their habitation with family members, participants were classified as “no” if they were living alone, and “yes” if they lived with family members. 

#### 2.2.5. Health-Related Factors

Average sleep time per day on weekdays, perceived stress, perceived health status, perceived body recognition, BMI, physical activity, alcohol consumption, depression, and smoking were selected as health-related factors [20]. Questions on alcohol consumption were addressed as part of the Alcohol Use Disorder Identification Test (WHO; AUDIT), which was used to screen for drinking risk [21]. AUDIT is a simple and effective method used to screen for unhealthy alcohol use, such as hazardous consumption and alcohol use disorder [21]. Alcohol use was scored using “frequency of drinking”, “typical quantity”, and “frequency of heavy drinking” to identify the occurrence of hazardous alcohol use [21]. The hazardous alcohol use domain consisted of three questions: “How often do you drink alcohol?” (0 = never; 1 = monthly or less; 2 = 2–4 times a month; 3 = 2–3 times a week; 4 = 4 or more times a week); “How many drinks containing alcohol do you have on a typical day when you are drinking?” (0 = 1 or 2; 1 = 3 or 4; 2 = 5 or 6; 3 = 7–9; 4 = 10 or more); and “How often do you have six or more drinks on one occasion?” (0 = never; 1 = less than monthly; 2 = monthly; 3 = weekly; 4 = daily or almost daily) [21]. A score of <8 was classified as low risk, and that of ≥8 was classified as high risk, according to the WHO guidelines [21]. 

#### 2.2.6. Disease-Related Factors

Disease-related factors, including hypertension, stroke, MI, AP, and DM, were selected based on a review of previous studies [4,9]. These items were classified into “yes” or “no” according to the presence or absence of a diagnosis. 

#### 2.2.7. Depression

The Korean version of patient health questionnaire-9 (PHQ-9) was used for the 2020 KNHANES [22]. Each item in the questionnaire was rated on a scale of 0–3 based on how much a symptom bothered the respondents (0 = not at all; 1 = several days; 2 = more than half a day; 3 = nearly every day), with the total score ranging from 0 to 27 [22]. A higher score indicated a higher level of depression, and the cut-off score was 10 [23]. Depression was classified as “yes” if the score was ≥10 and “no” if it was <10 [23].

### 2.3. Statistical Analysis

The sampling weights assigned to the participants were applied to all analyses to represent the Korean population and were considered a complex sample design; stratification was also conducted [20]. Analyses were performed with the IBM SPSS software (version 24.0; IBM Corp., Armonk, NY, USA). The Rao–Scott χ^2^ test was performed to evaluate differences in baseline characteristics based on OSA risk factors. A *t*-test was performed to evaluate differences in the baseline QoL based on OSA risk factors. The data were shown as weighted percentages and unweighted frequencies. Complex sample linear regression analysis was performed on weighted data using a complex sample procedure to identify the effect of the groups with and without the risk factors for OSA on the QoL. To assess differences in the QoL based on the risk severity of OSA, an analysis of differences was performed by controlling for variables that influenced the QoL of both groups. 

### 2.4. Ethical Consideration

The KNHANES was conducted with the approval of the Research Ethics Review Board of the KDCA (2018-01-03-2C-A). This study used the raw data published in the KNHANES after obtaining permission from the KDCA.

## 3. Results

The number of participants with MetS aged ≥ 40 years was 1589 in the 2020 KNHANES. There were 748 participants with low-risk factors for OSA, and 841 participants with intermediate risk factors or higher risk factors for OSA (Table 1). 

### 3.1. QoL

QoL recorded in the low-risk group was 0.966 ± 0.001, while QoL in the intermediate or higher risk group was 0.941 ± 0.004. The difference between the two groups in QoL was statistically significant (F = 56.322, *p* < 0.001) (Table 2). 

### 3.2. Sociodemographic Characteristics of the Participants According to Risk Factors for OSA

Among individuals with MetS, the QoL of the low-risk group for OSA showed differences based on gender (F = 20.171, *p* < 0.001), age (F = 23.056, *p* < 0.001), education (F = 71.460, *p* < 0.001), occupation (F = 23.620, *p* < 0.001), and household income (F = 15.803, *p* < 0.001). Similarly, the QoL of the high-risk group for OSA among individuals with MetS differed based on gender (F = 49.213, *p* < 0.001), age (F = 21.366, *p* < 0.001), education (F = 38.136, *p* < 0.001), living alone (F = 4.266, *p* = 0.041), occupation (F = 42.790, *p* < 0.001), and household income (F = 18.405, *p* < 0.001) (Table 3). 

### 3.3. Health- and Disease-Related Characteristics of Participants According to Risk Factors for OSA

The analysis of QoL differences was performed based on health and disease-related characteristics. In the low-risk group, QoL differences were observed in perceived health status (F = 32.279, *p* < 0.001), smoking (F = 6.941, *p* = 0.001), binge alcohol consumption (F = 5.600, *p* = 0.019), depression (F = 18.717, *p* < 0.001), hypertension (F = 5.314, *p* = 0.023), cerebrovascular accidents (CVA) (F = 5.042, *p* = 0.026), and diabetes mellitus (DM) (F = 11.393, *p* = 0.001). In the high-risk group, differences in QoL were observed in perceived stress (F = 6.64, *p* = 0.011), perceived health status (F = 32.542, *p* < 0.001), physical activity (F = 9.072, *p* = 0.003), smoking (F = 9.753, *p* < 0.001), binge alcohol consumption (F = 7.441, *p* = 0.007), depression (F = 35.842, *p* < 0.001), hypertension (F = 12.729, *p* < 0.001), CVA (F = 9.066, *p* = 0.003), and DM (F = 16.104, *p* < 0.001) (Table 4).

### 3.4. Factors Influencing QoL by Group

Variables that showed differences between the low-risk group and high-risk group were included in the regression analysis. 

The variables that showed differences in QoL within the low-risk group were controlled as covariates. Subsequently, a multiple linear regression analysis was performed to identify factors associated with QoL. The results revealed that in the low-risk group, factors, such as age between 60 and 69 (*t* = 2.755, *p* = 0.007), education level below middle school (*t* = −2.565, *p* = 0.012), and lower levels of depression (*t* = 2.165, *p* = 0.033) were associated with QoL. Similarly, in the high-risk group, a regression analysis was performed by controlling for variables that showed differences in QoL. The results indicated that in the high-risk group, factors such as physical activity (*t* = 2.878, *p* = 0.005) and depression (*t* = 4.470, *p* < 0.001) were associated with QoL (Table 5). 

## 4. Discussion

This study was based on the following research questions: (i) Is there a difference in QoL between the group of patients with metabolic syndrome without risk factors for obstructive sleep apnea (OSA) and the group with risk factors for OSA? (ii) Are the factors influencing the QoL of these two groups the same or different? 

The group with a high risk of OSA had a lower QoL than that of the low-risk group, and differences were found in terms of QoL between the low- and high-risk groups (F = 56.332, *p* < 0.001). To identify independent variables related to the QoL of each group, the analysis controlled for variables that showed statistical differences. Subsequently, factors influencing the QoL of each group were examined separately. The results indicated that depression was a common factor influencing the QoL of both groups with low and high risks for OSA. In the low-risk group, an age between 60 and 69 and a lower level of education were factors related to QoL. Further, physical activity was identified as a factor associated with the QoL of the high-risk group. 

These findings align with those of Kim’s research, which identified age and depression as factors influencing QoL among elderly individuals with MetS [15]. These findings are also consistent with those of Limon’s study, which demonstrated an association between MetS and depression [24]. Studies on the QoL of patients with OSA found depression to be associated with QoL [8,18]. 

Depression is a psychological response that occurs in various individuals, not only in those suffering from MetS and OSA. As mental health deteriorates, such as in the case of depression, the risk of sleep disorders tends to increase. Conversely, high-quality sleep can contribute to improving mental health [25]. It is thought that patients with OSA may experience chronic sleep disorders more significantly due to depression. The data collection period for this study is 2020, so it is presumed that the prevalence of depression among the subjects might have been higher due to COVID-19. This warrants further comparison in the future. Depression is also associated with various diseases, such as cardiovascular disease and Alzheimer’s [26,27]. Therefore, continued interest in depression among individuals living in the community is needed. 

In the group with a high risk of OSA, there was no association between age and QoL. However, in the low-risk OSA group, there was an association between the age group 60–69 years and QoL. QoL for diabetes patients is associated with age, and decreases as age increases [28]. The results of this study indicate that in both groups, the QoL decreased with increasing age. The difference in QoL between the age groups of 60–69 and 70 and above was observed to be significantly substantial. 

Generally, 60–69 are the ages at which one retires from work and starts a new life. If individuals do not have activity restrictions owing to physical or mental problems in this period, they can engage in various activities in the local community. This is because local senior welfare centers or administrative welfare centers conduct programs for seniors, such as table tennis, billiards, badminton, and singing classes. Therefore, compared to those over 70 years of age, ages 60–69 appear to be most related to QoL. 

In both the low-risk group and the high-risk group, QoL was higher in groups with higher education levels, in groups with a job compared to those without, and in groups with higher incomes compared to those with lower incomes. However, education emerged as a factor influencing QoL only in the low-risk group. Generally, lower educational attainment is associated with lower income, which is considered a determining factor of social economic status (SES) [29]. Additionally, education, occupation, and income are known to be key determinants of QoL. Income and SES determined via educational levels can contribute to social inequalities in health. Those with higher incomes can often afford to pay for health check-ups at hospitals with high service standards, allowing them to promptly detect and receive treatment for health issues. On the contrary, individuals with lower incomes may face challenges in accessing such be benefits. This holds true not only for patients with MetS but also for those with OSA [30]. SES linked to education, occupation, and income contributes to social disparities in sleep health [30]. 

This study’s results revealed that physical activity was a factor influencing QoL in the high-risk group. Physical activity reduces depression and stress and prevents obesity by increasing metabolic activity [31,32]. In this study, participants who engaged in physical activity made up 47.6% in the low-risk group and 41.4% in the high-risk group. The high-risk group in this study showed higher levels of obesity, depression, and stress compared to the low-risk group. Engaging in vigorous physical activities such as heavy lifting, digging, and aerobic exercises, as well as walking, has been shown to reduce the prevalence of OSA [33,34]. Physical activity improves quality of sleep and enhances QoL [35,36]. COVID-19 has led to difficulties in daily life, reducing physical activity and sleep while increasing feelings of depression [37]. Due to these temporal and environmental conditions, it is thought that physical activity in the high-risk group has emerged as a factor related to QoL. 

This study aimed to differentiate MetS subjects into groups with and without the risk of OSA, examining the factors influencing quality of life in each group. In the low-risk group, age, education, and depression emerged as factors influencing quality of life, whereas in the high-risk group, depression and physical activity played significant roles in influencing quality of life. However, it is important to acknowledge several limitations in this study. 

The limitation of this study is that it failed to consider various variables related to sleep, such as daytime sleepiness, the apnea–hypopnea index (AHI), and sleep quality. However, the study found that among the participants with MetS, the QoL of a group with risk factors for OSA and that of a group without risk factors for OSA were confirmed, and variables more closely related to their QoL were discovered.

## 5. Conclusions

This study compared the QoL of a group without risk factors for OSA with that of a group with risk factors for OSA among participants with MetS. The factors influencing the QoL of the low-risk group were age (60–69 years), education, and depression. The factors influencing the QoL of the high-risk group were physical activity and depression. These results suggest that the management of depression and physical activity is more crucial in the high-risk group. 

## Figures and Tables

**Figure 1 behavsci-14-00127-f001:**
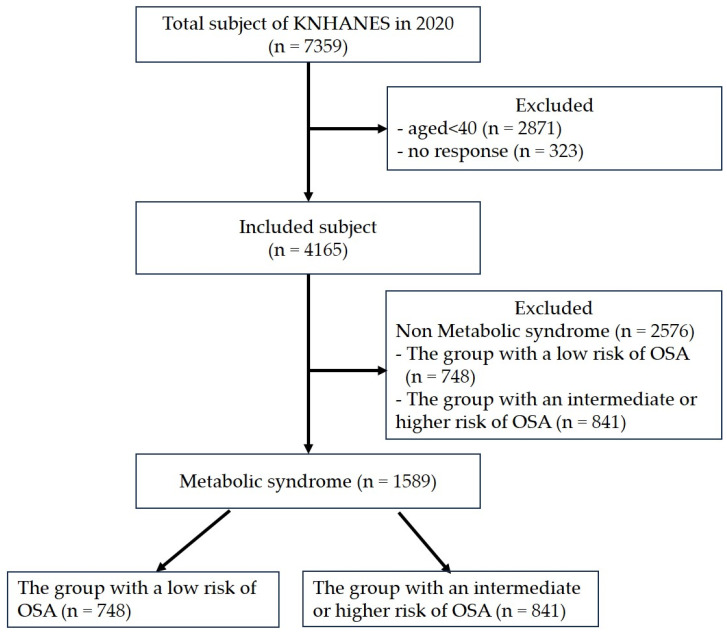
Flow diagram of the study participants.

**Table 1 behavsci-14-00127-t001:** STOP-Bang questionnaire.

(i) S (snoring)	Do you snore loudly (louder than talking or loud enough to be heard through closed doors)?
(ii) T (tired)	Do you often feel tired, fatigued, or sleepy during daytime?
(iii) O (observed)	Has anyone observed you stop breathing during sleep?
(iv) P (pressure)	Diastolic blood pressure of 90 mmHg or higher, systolic blood pressure of 140 mmHg, or taking antihypertensive medication.
(v) B (body mass index)	a BMI of more than 35 kg/m^2^
(vi) A (age)	Aged older than 50 years?
(vii) N (neck circumference)	40 cm or larger
(viii) G (gender)	Male

**Table 2 behavsci-14-00127-t002:** Differences in independent variables according to risk factors of OSA.

	MetS: Yes (n = 1589)			
Independent Variables	Low Risk Group (n = 748)	High Risk Group (n = 841)	F	*p*
	M (SE)	M (SE)		
EQ5D	0.966 (0.001)	0.941 (0.004)	56.322	<0.001

**Table 3 behavsci-14-00127-t003:** Sociodemographic characteristics of the participants according to risk factors for OSA.

Variables	Categories	Low-Risk Group (n = 748)			High-Risk Group (n = 841)			χ^2^ (*p*)
		n * (%) ^†^	M (SE)	F (*p*)	n * (%) ^†^	M (SE)	F (*p*)	
Gender	Male	173 (31.3)	0.972 (0.007)	20.171 (<0.001)	606 (78.0)	0.961 (0.004)	49.213 (<0.001)	252.629 (<0.001)
	Female	575 (68.7)	0.934 (0.005)		235 (22.0)	0.877 (0.011)		
Age	40–49	148 (28.3)	0.979 (0.005)	23.056 (<0.001)	101 (17.2)	0.973 (0.006)	21.366 (<0.001)	22.833 (<0.001)
	50–59	131 (20.3)	0.967 (0.005)		253 (38.5)	0.967 (0.005)		
	60–69	205 (22.8)	0.946 (0.009)		258 (26.1)	0.922 (0.009)		
	≥70	264 (28.6)	0.883 (0.011)		229 (18.2)	0.879 (0.012)		
Education	<Middle school	309 (40.0)	0.903 (0.008)	71.460 (<0.001)	319 (32.7)	0.900 (0.009)	38.136 (<0.001)	7.213 (0.008)
	≥Middle school	318 (60.0)	0.975 (0.004)		467 (67.3)	0.961 (0.004)		
Living alone	Yes	160 (33.3)	0.916 (0.013)	0.732 (0.394)	132 (29.0)	0.891 (0.015)	4.266 (0.041)	1.060 (0.305)
	No	273 (66.7)	0.929 (0.008)		337 (71.0)	0.924 (0.008)		
Occupation	Yes	333 (57.6)	0.966 (0.005)	23.620 (<0.001)	457 (65.0)	0.964 (0.004)	42.790 (<0.001)	6.970 (0.009)
	No	294 (42.4)	0.919 (0.008)		329 (35.0)	0.898 (0.009)		
Household	Low	218 (33.7)	0.888 (0.011)	15.803 (<0.001)	201 (26.6)	0.866 (0.014)	18.405 (<0.001)	2.706 (0.070)
income	Middle	189 (33.5)	0.955 (0.007)		213 (34.8)	0.944 (0.007)		
	High	156 (32.8)	0.960 (0.007)		217 (38.6)	0.959 (0.006)		

* n is the non-weighted value; ^†^ % is the weighted value to correct for the target population.

**Table 4 behavsci-14-00127-t004:** Health- and disease-related characteristics of the participants according to risk factors for OSA.

Variables	Categories	Low Risk Group (n = 748)			High-Risk Group (n = 841)			χ^2^ (*p*)
		n * (%) ^†^	M (SE)	F (*p*)	n * (%) ^†^	M (SE)	F (*p*)	
Perceived stress	A little	576 (79.9)	0.950 (0.005)	3.105 (0.080)	615 (72.1)	0.948 (0.005)	6.64 (0.011)	8.559 (0.004)
	Much more	161 (20.1)	0.931 (0.009)		218 (27.9)	0.923 (0.009)		
Perceived health status	Bad	137 (18.8)	0.800 (0.011)	32.279 (<0.001)	236 (28.8)	0.888 (0.009)	32.542 (<0.001)	7.914 (<0.001)
	Moderate	340 (43.7)	0.954 (0.005)		401 (56.3)	0.954 (0.005)		
	Good	158 (26.9)	0.976 (0.005)		155 (20.6)	0.982 (0.005)		
BMI	Normal	304 (40.9)	0.943 (0.007)	1.348 (0.248)	232 (26.3)	0.939 (0.009)	0.353 (0.553)	28.170 (<0.001)
	Obesity	426 (59.1)	0.954 (0.005)		593 (73.7)	0.944 (0.005)		
Physical activity	Yes	298 (47.6)	0.950 (0.006)	0.448 (0.504)	332 (41.4)	0.955 (0.005)	9.072 (0.003)	3.379 (0.068)
	No	331 (52.4)	0.943 (0.006)		455 (58.6)	0.931 (0.006)		
Smoking	Current	80 (13.9)	0.968 (0.008)	6.941 (0.001)	189 (26.9)	0.957 (0.008)	9.753 (<0.001)	55.156 (<0.001)
	Past	114 (18.9)	0.964 (0.007)		320 (38.1)	0.954 (0.005)		
	Never	544 (67.2)	0.936 (0.005)		325 (35.0)	0.916 (0.008)		
Sleep duration	<7	337 (45.3)	0.944 (0.005)	0.596 (0.442)	403 (46.3)	0.938 (0.006)	0.460 (0.499)	0.105 (0.747)
	≥7	410 (54.7)	0.950 (0.006)		436 (53.7)	0.944 (0.006)		
ADIT	Low risk	316 (74.0)	0.952 (0.005)	5.600 (0.019)	340 (54.0)	0.942 (0.006)	7.441 (0.007)	29.439 (<0.001)
	High risk	83 (26.0)	0.972 (0.007)		241 (46.0)	0.964 (0.006)		
Depression	<10	604 (98.3)	0.951 (0.004)	18.717 (<0.001)	730 (94.5)	0.950 (0.004)	35.842 (<0.001)	18.119 (<0.001)
	≥10	17 (1.7)	0.742 (0.048)		52 (5.5)	0.775 (0.029)		
Hypertension	No	410 (70.4)	0.932 (0.006)	5.314 (0.023)	144 (29.6)	0.958 (0.005)	12.729 (<0.001)	170.411 (<0.001)
	Yes	338 (28.5)	0.955 (0.005)		697 (71.5)	0.929 (0.006)		
CVA	No	729 (98.1)	0.948 (0.004)	5.042 (0.026)	812 (97.2)	0.944 (0.004)	9.066 (0.003)	1.326 (0.251)
	Yes	19 (1.9)	0.862 (0.038)		29 (2.8)	0.837 (0.035)		
MI or AP	No	722 (97.0)	0.948 (0.004)	3.583 (0.060)	785 (94.8)	0.942 (0.005)	2.356 (0.127)	3.394 (0.067)
	Yes	26 (3.0)	0.894 (0.029)		56 (5.2)	0.921 (0.013)		
DM	No	564 (77.3)	0.955 (0.004)	11.393 (0.001)	606 (74.1)	0.951 (0.005)	16.104 (<0.001)	1.553 (0.215)
	Yes	184 (22.7)	0.911 (0.012)		235 (25.9)	0.912 (0.009)		

* n is the non-weighted value; ^†^ % is the weighted value to correct for the target population.

**Table 5 behavsci-14-00127-t005:** Factors influencing QoL by group.

Variables	Categories	Low-Risk Group (n = 748)			High-Risk Group (n = 841)		
		B	*t*	*p*	B	*T*	*p*
Gender	Male	0.025	1.962	0.057	0.024	0.806	0.422
	Female (ref.)						
Age	40–49	0.034	1.646	0.103	0.050	1.378	0.172
	50–59	0.035	1.768	0.080	0.025	1.252	0.214
	60–69	0.049	2.755	0.007	−0.022	−0.856	0.394
	70 (ref)						
Education	<Middle school	−0.029	−2.565	0.012	−0.018	−0.974	0.332
	≥Middle school (ref.)						
Living alone	Yes				−0.028	−1.657	0.101
	No						
Occupation	Yes	0.017	1.199	0.233	0.013	0.661	0.510
	No(ref)						
Household	Low	−0.017	−1.021	0.310	−0.011	−0.539	0.591
income	Middle	−0.002	−0.137	0.891	0.010	0.586	0.559
	High (ref.)						
AUDIT	Low risk	0.022	1.572	0.119	−0.024	−1.326	0.188
	High risk (ref.)						
Smoking	Current	0.011	0.816	0.417	−0.036	−1.479	0.143
	Past	0.006	0.455	0.650	−0.022	−0.832	0.408
	Never (ref.)						
Physical activity	Yes				0.050	2.878	0.005
	No						
Perceived stress	A little				−0.005	−0.277	0.783
	Much more (ref.)						
Perceived health status	Bad	−0.063	−3.387		−0.056	−1.978	0.051
	Moderate	−0.029	−2.429		0.007	0.272	0.786
	Good (ref.)						
Depression	<10	0.135	2.165	0.033	0.151	4.470	<0.001
	≥10 (ref.)						
Hypertension	No	0.001	0.045	0.964	0.004	0.291	0.771
	Yes (ref.)						
CVA	No	0.038	0.814	0.418	0.075	1.687	0.095
	Yes (ref.)						
DM	No	0.019	0.985	0.327	−0.022	−0.936	0.352
	Yes (ref.)						
		R^2^ = 0.300, F = 4.846, *p* < 0.001			R^2^ = 0.338, F = 9.347, *p* < 0.001		

## Data Availability

The datasets used in this study were obtained from https://knhanes.kdca.go.kr/knhanes/sub03/sub03_02_05.do (accessed on 23 February 2022).

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
