# Peer review of "Quality of Life in Metabolic Syndrome Patients Based on the Risk of Obstructive Sleep Apnea"

_behavsci, 2024, doi:10.3390/bs14020127_

Round 1

Reviewer 1 Report (Previous Reviewer 3)

Comments and Suggestions for Authors

·       Summary: This study about quality of life (QoL) in metabolic syndrome patients based on the degree of obstructive sleep apnea risk is an interesting topic. The overall QoL was not different between the two groups, which was statistically insignificant. Moreover, it demonstrated that age, education level and depression influence the QoL of the low-risk group. Depression also impacts the high-risk group in addition to physical activity.

·       Major concerns:

- According to Bianca Pivetta et al., the STOP-Bang questionnaire can be used as a screening tool in different geographical regions for stratifying OSA risk, except for East Asia because of craniofacial features. (Pivetta B, Chen L, Nagappa M, et al. Use and Performance of the STOP-Bang Questionnaire for Obstructive Sleep Apnea Screening Across Geographic Regions: A Systematic Review and Meta-Analysis. JAMA Netw Open. 2021;4(3):e211009. Published 2021 Mar 1). Therefore, I think when applying the STOP-Bang questionnaire in this study, additional anthropometric metrics are required for subjects who have more than two “yes” answers due to nonspecific symptoms of OSA included questionnaire (E.g., tiredness, high blood pressure, male) to increase the accuracy of OSA risk category.

- The second problem is age distribution in both two groups. In the low-risk group, the percentage of subjects over 60 years old is greater than the high-risk group ( 51.4% for the former and 44,3% for the latter). It may lead to analyzing incorrectly.

- The third problem is that physical activity needs to be more detailed in order to conclude if it influences the QoL of the high-risk group.

·       Minor concerns:

- The paragraph from line 115 to line 128 needs to be clarified to read. First, it does not explain what “three items” means. Second, I think  STOP-Bang parameters should be separated by using a table to increase the readability.

- The content of the paragraph from line 68 to line 72 overlaps the purpose section.

Author Response

Please see the attachment (PDF).

The formatting of the Word file is incorrect

Reviewer 2 Report (Previous Reviewer 2)

Comments and Suggestions for Authors

Interesting research authors made, it was also well written.

Authors should delve deeper into the multifaceted impact of demographic factors and socioeconomic indicators, including income, marital status, neighborhood and ethnicity. A comprehensive exploration of these variables is crucial for a nuanced understanding of OSA risk within diverse populations. Authors should explain more how their findings supports recents research confirming the influence of socioeconomic status, as reflected in income levels, on QOL which include overall well-being, thereby contributing to OSA risk and related disease like excessive daytime sleepiness.

Marital status may also play a role, with potential variations in sleep patterns and environments. Analyzing neighborhood characteristics and their association with OSA risk can uncover environmental factors, such as air quality or noise pollution, that may exacerbate or mitigate the living conditions of people without OSA. Furthermore, a thoughtful discussion on ethnicity should consider genetic predispositions, cultural practices and healthcare disparities that may contribute to varying OSA prevalence among different ethnic groups; knowing that all of them affects Mets regardless OSA risk. By thoroughly examining these socio-demographic dimensions, authors can provide a more comprehensive perspective on the social determinants of OSA and contribute valuable insights for healthcare interventions and policy considerations.

References suggestions for the discussions (and may be results) update are the following:

1-Socioeconomic Position and Excessive Daytime Sleepiness: A Systematic Review of Social Epidemiological Studies.

2-The influence of job stress, social support and health status on intermittent and chronic sleep disturbance: an 8-year longitudinal analysis

3-Work environment mediates a large part of social inequalities in the incidence of several common cardiovascular risk factors: Findings from the Gazel cohort

4-Sleep duration and the associated cardiometabolic risk scores in adults

5-Social conditions as fundamental causes of health inequalities: theory, evidence, and policy implications

6-Social disparities in sleep health of African populations: A systematic review and meta-analysis of observational studies

7-Relationship between sleep duration and body mass index depends on age

Comments on the Quality of English Language

Minor editing of English language required

Author Response

Please see the attachment (PDF).

The formatting of the Word file is incorrect

Round 2

Reviewer 1 Report (Previous Reviewer 3)

Comments and Suggestions for Authors

The manuscript has been revised well.

Reviewer 2 Report (Previous Reviewer 2)

Comments and Suggestions for Authors

This revised version looks better than the previous, with a more balanced and convincing discussion. Thank to research team for their efforts, i endorse thsi version.

This manuscript is a resubmission of an earlier submission. The following is a list of the peer review reports and author responses from that submission.

Round 1

Reviewer 1 Report

Comments and Suggestions for Authors

This manuscript is an analysis of the Korea National Health and Nutrition Examination Survey (KNHANES), a nationwide cross-sectional survey conducted annually by the Korean Disease Control and Prevention Agency (KDCA).

It is a very interesting and well-written paper.

I recommend its publication after some minor improvements:

-       Add a paragraph on limitations of the study at the end of discussion.

-       In conclusions, line 286 is confusing. Please explain in this line the two groups.

Reviewer 2 Report

Comments and Suggestions for Authors

I have read with attention your paper and it is a good contribution. Methodology is clear and analysis are well explained. It will be interesting to write a paragraph explaining the influence of socioeconomic status (SES), due to the demographic variables you collected.

Understanding the etiology of socioeconomic disparities in health could assist public health authorities in preventing the morbidity of socially disadvantaged individuals. It is particularly true when a sleep disturbance is involved. A recent systematic review provided evidence that SES and OSA are strongly associated. Their findings were coherent with dothers studies reaching similar conclusions with differents diseases in different countries. Another highly cited papers even showed that such a relation can be measured objectively with actigraphy and polysomnography. To my opinion, discussed how income, education or neighborhood could affect interesting results you have, may continue debate about the consideration of social and environmental factors in biomedical research. I am looking forward to see your arguments.

Here below some articles useful for your writing:

1-Objective and subjective socioeconomic gradients exist for sleep in children and adolescents. Health Psychol. 2014 Mar;33(3):301-5. doi: 10.1037/a0032924.

2-Childhood sleep apnea and neighbourhood disadvantage. J. Pediatr. 2011, 158, 789–795.e1

3-Socioeconomic Position and Excessive Daytime Sleepiness: A Systematic Review of Social Epidemiological Studies. Clocks & Sleep 2022

4-Social inequalities in sleep-disordered breathing: Evidence from the CoLaus|HypnoLaus study. J. Sleep Res. 2019

Comments on the Quality of English Language

Some sentences can be shorten.

Reviewer 3 Report

Comments and Suggestions for Authors

This study about quality of life in metabolic syndrome patients based on the risk of obstructive sleep apnea is an exciting topic. It shows that the QoL is correlated with the degree of OSA’s risks. Moreover, some psychological factors affecting the QoL included the perceived health status and stress. However, there are several concerns in this study.

Major point

1)     The population size in two groups; Metabolic syndrome group and non-Metabolic syndrome group are unequal. Because this will cause some problems when comparing them. To compare these two groups correctly, adjustment by gender, age, and education should be considered.

2)     This study lacks the sleep studies of the patients. In intermediate and high risk of OSA, I think the parameter of PSG (e.g., AHI) needs to be mentioned because OSA may influence these findings in this group.

3)     The risk score of sleep apnea includes fatigue. Therefore, it seems to be difficult to exclude the patients who have just depression from the OSA risk group in this study design.   

4)     This article should add the flowchart of research and charts to improve the readability.

Minor points

1)     none
